

# Manure application increased denitrifying gene abundance in a drip-irrigated cotton field

Mingyuan Yin[1,2,3], Xiaopeng Gao[1,2,4], Mario Tenuta[4], Wennong Kuang[1,2,3], Dongwei Gui[1,2] and Fanjiang Zeng[1,2]

[1] State Key Laboratory of Desert and Oasis Ecology, Xinjiang Institute of Ecology and Geography, Chinese Academy of Sciences, Urumqi, China
[2] Cele National Station of Observation and Research for Desert-Grassland Ecosystem, Xinjiang Institute of Ecology and Geography, Chinese Academy of Sciences, Cele, China
[3] University of Chinese Academy of Sciences, Beijing, China
[4] Department of Soil Science, University of Manitoba, Winnipeg, MB, Canada

Corresponding author
Xiaopeng Gao,
xiaopeng.gao@umanitoba.ca

## ABSTRACT

Application of inorganic nitrogen (N) fertilizer and manure can increase nitrous oxide ($N_2O$) emissions. We tested the hypothesis that increased $N_2O$ flux from soils amended with manure reflects a change in bacterial community structure and, specifically, an increase in the number of denitrifiers. To test this hypothesis, a field experiment was conducted in a drip-irrigated cotton field in an arid region of northwestern China. Treatments included plots that were not amended (Control), and plots amended with urea (Urea), animal manure (Manure) and a 50/50 mix of urea and manure (U+M). Manure was broadcast-incorporated into the soil before seeding while urea was split-applied with drip irrigation (fertigation) over the growing season. The addition treatments did not, as assessed by nextgen sequencing of PCR-amplicons generated from rRNA genes in soil, affect the alpha diversity of bacterial communities but did change the beta diversity. Compared to the Control, the addition of manure (U+M and Manure) significantly increased the abundance of genes associated with nitrate reduction (*narG*) and denitrfication (*nirK* and *nosZ*). Manure addition (U+M and Manure) did not affect the nitrifying enzyme activity (NEA) of soil but resulted in 39–59 times greater denitrifying enzyme activity (DEA). In contrast, urea application had no impact on the abundances of nitrifier and denitrifier genes, DEA and NEA; likely due to a limitation of C availability. DEA was highly correlated ($r = 0.70$–$0.84$, $P < 0.01$) with the abundance of genes *narG*, *nirK* and *nosZ*. An increase in the abundance of these functional genes was further correlated with soil $NO_3^-$, dissolved organic carbon, total C, and total N concentrations, and soil C:N ratio. These results demonstrated a positive relationship between the abundances of denitrifying functional genes (*narG*, *nirK* and *nosZ*) and denitrification potential, suggesting that manure application increased $N_2O$ emission by increasing denitrification and the population of bacteria that mediated that process.

## INTRODUCTION

Nitrous oxide ($N_2O$) accounts for nearly 8% of the warming impact of anthropogenic activities and contributes to the depletion of ozone in the stratosphere (*Ravishankara, Daniel & Portmann, 2009*). The $N_2O$ concentration in the atmosphere has increased at a rate of 0.26% per year, with more than 80% of the emissions associated with agricultural activities where organic (e.g., animal manures) or inorganic (e.g., synthetic fertilizers) sources of nitrogen (N) are added to soil (*IPCC, 2013*). Manure application can result in more $N_2O$ emissions than inorganic N fertilizers (*Watanabe et al., 2014*; *Zhou et al., 2017*), which we also observed in a drip-irrigated cotton field with low soil organic carbon in arid northwestern China (*Kuang et al., 2018*). However, it remains unclear whether the increased emissions with manure are linked with changes in the microbial community, especially those involved in the processes of nitrification and denitrification.

Nitrification is a biological oxidation process in which ammonia is converted to nitrate via nitrite ($NH_3 \rightarrow NH_2OH/HNO \rightarrow NO_2^- \rightarrow NO_3^-$). The steps of nitrification are controlled by nitrifier functional genes, including (1) ammonia-oxidizing bacterial (*AOB*) and (2) archaea (*AOA*) genes, and (3) nitrite-oxidizing bacterial genes. The first step in the oxidation of ammonia to $NH_2OH$ limits the entire nitrification reaction (*Kowalchuk & Stephen, 2001*). Applications of manure or inorganic N can exert a significant impact on nitrification. For example, *Tao et al. (2017)* reported that fertilizer N drives the abundance, community structure, and activity of nitrifying bacteria. Long-term application of manure and inorganic fertilizers reduced the copy number of *AOA* but increased that of *AOB* for agricultural soils in the cold climate of northeast China (*Fan et al., 2011*). For desert topsoil in Arizona, USA, long-term inorganic N addition did not affect the community structure of ammonia-oxidizing microorganisms but increased the *amoA* gene abundance of both *AOA* and *AOB* (*Marusenko, Garcia-Pichel & Hall, 2015*). In contrast, a recent study of fertilized subtropical forest soils in southern China found that soil factors such as $NH_4^+$ concentration and pH-controlled nitrification and denitrification activities, rather than the abundance and community structure of N-cycling prokaryotes (*Tang et al., 2019*).

Denitrification is a multi-step reduction process of $NO_3^-$ to $N_2$ ($NO_3^- \rightarrow NO_2^- \rightarrow NO \rightarrow N_2O \rightarrow N_2$) mediated mainly by many bacteria under limited oxygen conditions. Specific reductases encoded by functional genes regulate each step of the reaction, including, nitrate reductase (e.g., *narG*, *napA*), nitrite reductase (e.g., *nirS*, *nirK*), nitric oxide reductase (e.g., *cnorB*, *qnorB*) and nitrous oxide reductase (*nosZ*; *Simon & Klotz, 2013*). Changes in the abundance and community structure of denitrifiers can largely explain the increase in denitrification associated with fertilizer application (*Yin et al., 2015*). In a 160-year-long field experiment, *Clark et al. (2012)* reported long-term manure application increased denitrification compared to inorganic N fertilizer addition, which was mainly attributed to increased abundance of *nirK*- but not *nirS*-type denitrifiers. In contrast, several other studies reported that soil properties such as soil water content

and N availability other than the abundance of denitrifier, deterimined the rate of denitrification (*Attard et al., 2011*; *Shrewsbury et al., 2016*).

Nitrogen additions can affect soil microbial community directly by supplying substrates for microorganisms or indirectly by changing soil properties. Application of animal manure can increase microbial biomass and diversity by providing carbon sources for microorganisms. In contrast, inorganic N application generally reduces soil microbial community diversity. For example, *Zhang et al. (2017)* recently reported that the application of inorganic fertilizers to acidic and near-neutral soils in a maize-vegetable rotation in southwest China significantly reduced bacterial diversity. *Sun et al. (2015)* also reported that the application of inorganic fertilizer to a wheat-soybean rotation for 30 years in central China reduced soil bacterial richness and diversity. Application of inorganic fertilizer affected the soil microbial community mainly by decreasing soil pH (*Geisseler & Scow, 2014*).

As a dominant cash crop in northwestern China, cotton receives intensive inputs of inorganic fertilizers and more recently, water as drip-irrigation (*Dai & Dong, 2014*). Cattle and sheep manure are also often used as nutrient sources due to their availabilities from local livestock production. In this region, manure application can greatly increase $N_2O$ emissions compared with granular urea, although emissions under drip irrigation were generally low (*Kuang et al., 2018*; *Ma et al., 2018*). Both nitrification and denitrification could play a role in production and emission of $N_2O$ under field conditions. It remains unclear how additions of organic manure or inorganic fertilizer affect the gene abundances and activity of nitrifier and denitrifier communities under drip irrigated conditions.

The objective of this study was to determine the influence of inorganic fertilizer and manure application to soil on the abundance and activities of $N_2O$-producing functional genes, as well as bacterial community structure in a drip-irrigated cotton field. We hypothesized that $N_2O$ emissions would increase with manure application and the abundance and activity of denitrifiers and not nitrifiers.

## MATERIALS AND METHODS

### Site description and experimental design

A plot-based field experiment was conducted at the Cele Research Station (37°01′N, 80°43′E) of the Chinese Academy of Sciences in the 2015 and 2016 summer growing seasons. The region has a typically arid continental climate with an extremely low long-term average annual precipitation of only 42 mm, mainly distributed between May and July. The long-term average mean annual air temperature is 12.7 °C. The soil is classified as Aridisols in the USDA soil taxonomy system or Gypsisol in the FAO World Reference Base for Soil Resources. At the start of the study, the surface soil (0–20 cm) was a fine sand texture (sand 90%, silt 4%, clay 6%) with bulk density 1.46 Mg m$^{-3}$, pH$_{H2O}$ 8.0, electrical conductivity (EC) 144.4 μS cm$^{-1}$, total Kjeldahl N 0.31 g kg$^{-1}$, extractable $NO_3^-$-N 25.7 mg kg$^{-1}$, 0.5M NaHCO$_3$-extractable P 14.6 mg kg$^{-1}$, 1.0M ammonium acetate K 153 mg kg$^{-1}$ and organic matter 6.9 g kg$^{-1}$. Analysis of soil characteristics was based on *Carter (1993)*. Before this study, the experimental field was cropped to cotton for over

5 years and received both manure and urea applications each year, in accordance with local farmer practice.

The experimental design was previously described in *Kuang et al. (2018)* and only treatments under drip irrigation used in the current study. Briefly, the study was a randomized complete block design of four treatments with four replicate plots for a total of 16 plots. Each plot was 10 m long × 6 m wide and was separated from the other plots by a 1.1-m non-cropped buffer area. Treatments included (1) an unfertilized control (Control), and application of 240 kg of available N ha$^{-1}$ in the form of (2) granular urea (Urea, 46-0-0), (3) mixture of sheep and cattle manure compost (Manure), and (4) 50% urea with 50% manure by weight (U+M). Local producers commonly use such N application rates for high-yielding cotton fields. For the Urea treatment, 20% of added N was banded in the plant row before planting, and the rest was applied with irrigation water as a schedule of 5% at 9 weeks, and 15% of added N each at 11, 14, 15, 16 and 17 weeks after planting. The manure was all applied before planting by broadcast-incorporation at 10 cm depth. The manure had a moisture content of 25% and a dry weight-based total N, P, K content of 15.6, 2.0 and 16.8 g kg$^{-1}$, respectively. Analysis of manure was done on subsamples digested with a mixture of perchloric, sulfuric and hydrofluoric acid. Total P and K in the acid digestion were measured using the Mo-Sb colorimetric method and atomic absorption spectrometry (ICE3500; Thermo Fisher, Waltham, MA, USA), respectively. Total N was determined colorimetrically after Kjeldahl digestion. The manure had an available N concentration of 29 mg N kg$^{-1}$, determined by the alkaline hydrolyze method. In each year, cottonseed (c.v. Xinluzao 48; Huiyuan Tech, Shihezi, China) was planted at 75 kg ha$^{-1}$ in early to middle April under plastic-mulch and drip-irrigated, which is typical of recent cotton production in the region. Details on the system were described by *Kuang et al. (2018)*. Before seeding, all plots received a broadcast-incorporated application of 120 kg P$_2$O$_5$ ha$^{-1}$ as calcium phosphate and 60 kg K$_2$O ha$^{-1}$ as K$_2$SO$_4$.

## Soil sampling

Soil samples (0–20 cm depth) were collected with a hand auger (2.5 cm diameter) in September 2016 with cotton at the boll opening stage. For each plot, four soil cores were collected next to the drip tape and mixed thoroughly together for one composite sample per plot. The auger was cleaned using 95% alcohol and wiped with sterile paper before collecting the next soil sample. Each sample was passed through a two mm mesh screen and partitioned into three subsamples. One subsample was air-dried at room temperature for chemical analysis. The second subsample for analysis of denitrifying enzyme activity (DEA) and nitrifying enzyme activity (NEA) was stored at −20 °C and analyzed within 1 week. The third subsample for microbial molecular analysis was stored at −80 °C.

## Soil chemical properties

Soil NH$_4^+$ and NO$_3^-$ was extracted using 0.01M CaCl$_2$ and measured with a continuous flow analyzer (SEAL Analytical, Norderstedt, Germany). Soil pH was measured at 1:2.5 soil:water ratio. Soil total C was measured using by wet oxidation method with potassium

dichromate. Total N was analyzed by Kjeldahl acid-digestion method with a Kjeltec 1035 analyzer (Tecator AB; Sweden). Available Fe and Cu were extracted with diethylenetriamine pentaacetic acid (DPTA) (0.005M DPTA + 0.1M triethanolamine + 0.01M $CaCl_2$ set to pH 7.3) and analyzed using ICP-OES (730ES; VARIAN, Palo Alto, CA, USA). Soil dissolved organic carbon (DOC) was extracted using deionized water (1:5 soil:water ratio) and analyzed using a TOC analyzer (Aurora 1030W; OI Analytical, College Station, CA, USA). Soil C:N ratio was calculated on mass basis of total C and total N.

## Determination of denitrifying and nitrifying enzyme activity

The frozen soil samples were pre-incubated to thaw at 25 °C for 2 days before analysis of DEA and NEA. Soil DEA was expressed as the rate of $N_2O$ production ($\mu$g N $h^{-1}$ $g^{-1}$ soil) and determined using the anaerobic slurry technique (*Beauchamp & Bergstrom, 1993*). Briefly, 25 g thawed soil samples were placed into 125 ml plasma flasks. A total of 25 ml solution including 10 mM $KNO_3$, 10 mM glucose, 50 mM $K_2HPO_4$ and 0.1 g $L^{-1}$ chloramphenicol to inhibit new protein production was added to each flask. The flasks were evacuated and flushed with a 90:10 He-$C_2H_2$ gas mixture to create anaerobic conditions and suppress $N_2O$-reductase activity. Flasks were then shaken for 60 min and gas samples taken 0, 15, 30, 45 and 60 min after the onset of mixing using an orbital shake (180 rpm). Concentrations of $N_2O$ in gas samples were immediately analyzed using gas chromatography equipped with an electron capture detector (Agilent 7890A; Agilent Technologies, Santa Clara, CA, USA).

Soil NEA was expressed as $\mu$g $NO_3^-$-N $h^{-1}$ $g^{-1}$ dry soil and determined according to *Hart et al. (1994)*. Briefly, a thawed soil sample (15 g dry soil equivalent) was placed into a 250 ml plasma flask with 100 ml solution of 1.5 mM $(NH_4)_2SO_4$ and one mM phosphate buffer (pH = 7.2). The flask was incubated at room temperature under constant agitation (180 rpm). Samples of the slurry were taken at 2, 4, 8, 12 and 24 h during incubation. Concentrations of $NO_2^-$ and $NO_3^-$ in the samples was then determined using the continuous flow analyzer. NEA rate was calculated from the slope of the regression model of $NO_2^-$ plus $NO_3^-$ concentrations and time.

## Soil DNA extraction and real-time PCR

Soil DNA was extracted from 0.3 g of a soil sample using the Power Soil Total DNA Isolation Kit (MO-BIO Laboratories Inc., Carlsbad, CA, USA) according to the manufacturer's instructions. The quality and concentration of DNA were estimated using a Nanodrop 1000 Spectrophotometer (Thermo Fisher, Waltham, MA, USA) and gel electrophoresis (1.0% agarose). The DNA extracts were diluted at a ratio of 1:10 with double-distilled water (ddH$_2$O) to reduce the potential for PCR inhibition and then stored at −20 °C until use.

Quantitative PCR was used to quantify archaeal *amoA* and bacterial *amoA, narG, nirK, nirS* and *nosZ* gene in triplicate. All reactions were carried out using a CFX96$^{TM}$ (BIO-RAD, Laboratories Inc., Hercules, CA, USA). Each PCR reaction mixture contained one $\mu$l of 10-fold diluted soil DNA as a template, 10 $\mu$l SYBR$^{®}$ Premix Ex Taq$^{TM}$ II

**Table 1 The primer sets and thermocycling conditions used for quantitative PCR reactions.**

| Target gene | Primer set | Sequence (5′–3′) | Product size (bp) | Annealing time and temperature | Elongation time and temperature | Reference |
|---|---|---|---|---|---|---|
| Archaeal amoA | Arch-amoAF Arch-amoAR | STAATGGTCTGGCTTAGACG GCGGCCATCCATCTGTATGT | 635 | 30 s, 55 °C | 30 s, 72 °C | Francis et al. (2005) |
| Bacterial amoA | amoA1F amoA2R | GGGGTTTCTACTGGTGGT CCCCTCKGSAAAGCCTTCTTC | 491 | 30 s, 56 °C | 30 s, 72 °C | Rotthauwe, Witzel & Liesack (1997) |
| narG | narGG-F narGG-R | TCGCCSATYCCGGCSATGTC GAGTTGTACCAGTCRGCSGAYTCSG | 173 | 30 s, 55 °C | 30 s, 72 °C | Bru, Sarr & Philippot (2007) |
| nirS | nirS4QF nirS6QR | GTSAACGYSAAGGARACSGG GASTTCGGRTGSGTCTTSAYGA | 465 | 30 s, 60 °C | 30 s, 72 °C | Throback et al. (2004) |
| nirK | FlaCu R3Cu | ATCATGGTSCTGCCGCG GCCTCGATCAGRTTGTGGTT | 474 | 30 s, 63 °C | 30 s, 72 °C | Throback et al. (2004) |
| nosZ | nosZF nosZ-1622R | CGYTGTTCMTCGACAGCCG CGSACCTTSTTGCCSTYGCG | 453 | 30 s, 61 °C | 35 s, 72 °C | Scala & Kerkhof (1998) |

(TaKaRa, Kusatsu, Japan), 0.8 µl of primer (10 µM), and 7.4 µl ddH$_2$O in a total volume of 20 µl. Primers and thermocycling conditions used in the qPCR reactions are given in Table 1. Plasmids containing respective sequences of the targeted genes were generated by cloning the targeted gene fragments from soil DNA into plasmid pMD$^{TM}$ 19-T Vector (TaKaRa, Kusatsu, Japan). Standard curves for each gene were created from 10-fold serial dilutions ($10^8$–$10^1$) of the known quantities of linearized plasmid DNA harboring the target gene sequences. All qPCR reactions were conducted in triplicate. The qPCR efficiency and slope were 92% and −3.5 ($R^2 = 0.990$) for archaeal amoA, 105% and −3.2 ($R^2 = 0.999$) for bacterial amoA, 90% and −3.7 ($R^2 = 0.999$) for narG, 85% and −3.7 ($R^2 = 0.997$) for nirS, 96% and −3.4 ($R^2 = 0.998$) for nirK, and 80% and −3.5 ($R^2 = 0.990$) for nosZ, respectively. The generally low qPCR efficiency for nirS and nosZ genes agreed with previous studies reporting similar levels (74–90%, Ding et al., 2015; Harter et al., 2014).

## High-throughput sequencing

The 16S rRNA gene of the V3-V4 hypervariable region was analyzed by MiSeq sequencing on the Illumina Miseq 2 × 300 bp platform at Shanghai Sangon Biotech Co., Ltd. with the universal primers 515F (GTGCCAGCMGCCGCGG) and 907R (CCGTCAATTCMTTTRAGTTT) that amplify both bacteria and archaea DNA (Li et al., 2014). Both forward and reverse primers were added with a barcode. The thermocycling program was set as: an initial denaturation at 94 °C for 3 min, five cycles at 94 °C for 30 s, 45 °C for 20 s, 65 °C for 30 s of extension, then 20 cycles of 94 °C for 20 s, 55 °C for 20 s, 72 °C for 30 s, with a final extension at 72 °C for 5 min. The reactions were set as: 15 µl 2 × Taq Master Mix (Thermo Scientific, Waltham, MA, USA), two µl of DNA template (about 20 ng), one µl of each appropriate primer (10 µM), 11 µl of ddH$_2$O. The PCR products were purified and quantified using the Agencourt AMPure XP (Beckman Coulter, Brea, CA, USA) reagent and Qubit$^{TM}$ ssDNA Assay Kit (Life Technologies, Carlsbad, CA, USA), respectively. Finally, the purified PCR products of

each sample were equally combined based on their concentrations and produced a DNA pool which included 16S rRNA gene amplified fragments for sequencing.

Sequencing reads were allocated to each sample based on their unique barcodes. Raw sequences were firstly processed using cutadapt software (v 1.2.1) to trim the barcodes of their primers. Two short Illumina reads were then merged with PEAR (v 0.9.6) software (*Zhang et al., 2014*), and finally, PRINSEQ software (v 0.20.4, *Schmieder & Edwards, 2011*) was used for quality control of the merged reads. Only sequences >200 bp in length with an average quality score >40 were used for further analyses. Chimeras were filtered by comparing the sequences with those in the reference database using the UCHIME algorithm (v 4.2.40, *Edgar et al., 2011*). After the above screening, the remaining high-quality sequences were clustered into operational taxonomic units (OTUs) at a ≥97% similarity identity threshold. The singletons and low abundance OTUs were removed before further analyses. The Ribosomal Database Project classifier (*Wang et al., 2007*) was used to identify taxonomic information at the bootstrap cutoff of 80%. Based on the OTUs output, α-diversity, and β-diversity, and canonical correspondence analysis (CCA) were performed. Species richness and diversity indices, including coverage, Chao1, ACE, Simpson and Shannon were calculated using mothur (v 1.30.1, *Schloss et al., 2009*) to estimate the α-diversity of each sample.

## Statistical analysis

Treatment effects on soil properties, NEA, DEA, α-diversity indices and bacterial gene abundances were conducted using a one-way ANOVA. Pearson correlation analysis was conducted to assess the relationships between the functional gene abundances, NEA, DEA and selected soil properties. ANOVA and Pearson correlation analysis was performed with SAS 9.3 (SAS Institute, Cary, NC, USA) and differences were considered significant at $P < 0.05$. Principal coordinates analysis (PCoA) was performed to determine the community β-diversity of bacteria using the Vegan package Version 1.17-7 (*Oksanen, 2011*) implemented with the R language, which was based on a bacterial weighted UniFrac metric matrix. CCA was performed with the Vegan package implemented with the R language to determine the relationships between soil physicochemical properties and bacterial communities. Untransformed data were used for the PCoA and CCA analyses. The relative abundances of bacterial community at the phylum level between treatments were compared using the Welch's *t*-test with STAMP (Statistical Analysis of Metagenomic Profiles). Corrected *P*-values of the Welch's *t*-test were calculated using the false discovery rate for multiple testing correction.

## RESULTS

### Soil chemical characteristics

Manure and U+M treatments increased soil total N content by half compared to the unfertilized control (Table 2). Soil $NO_3^-$ concentrations with Manure and U+M treatments were 120 and 103 mg kg$^{-1}$, respectively, being 2.4–4.8 times greater than the Urea and Control treatments. In contrast, soil $NH_4^+$ concentrations were not affected by the treatments. Soil total C and DOC were also greater in Manure and U+M compared to the

**Table 2 Soil (0–20 cm) properties following addition of treatments in the drip-irrigated cotton field used in this study.**

| Treatment | Total N content (g kg$^{-1}$) | NO$_3^-$ (mg kg$^{-1}$) | NH$_4^+$ (mg kg$^{-1}$) | Total C content (g kg$^{-1}$) | DOC (mg g$^{-1}$) | C:N |
|---|---|---|---|---|---|---|
| Control | 0.9 ± 0.1[c] | 21 ± 3[b] | 14.1 ± 0.1[a] | 7.8 ± 0.5[b] | 0.21 ± 0.01[b] | 9.2 ± 0.7[bc] |
| Urea | 1.0 ± 0.2[bc] | 30 ± 2[b] | 17.9 ± 3.2[a] | 6.8 ± 0.4[b] | 0.20 ± 0.02[b] | 6.8 ± 1.3[c] |
| Manure | 1.3 ± 0.1[ab] | 120 ± 29[a] | 15.4 ± 1.5[a] | 15.9 ± 1.4[a] | 0.37 ± 0.04[a] | 12.6 ± 0.8[ab] |
| U+M | 1.4 ± 0.1[a] | 103 ± 7[a] | 17.9 ± 2.6[a] | 19.1 ± 1.8[a] | 0.36 ± 0.02[a] | 13.6 ± 1.3[a] |

Notes:
U+M: 50% urea + 50% manure. Values are the mean ± 1 standard error, $n = 4$.
Means within a column followed by the same letter are not significantly different at $P < 0.05$ (Tukey's HSD).

Control and Urea treatments. As a result, treatments with manure addition (Manure and U+M) had 37–100% higher soil C:N ratios, compared to Urea and Control.

## Denitrifying enzyme activity and nitrifying enzyme activity

Manure and U+M treatment significantly ($P < 0.001$) increased DEA levels compared to the Urea and Control treatments (Fig. 1A). In contrast, NEA levels did not respond significantly to any amendment treatment in spite of an increasing trend from control to urea and manure additions (Fig. 1B).

## Bacterial community, and nitrifier and denitrifier gene abundances

The sequence coverage index ranged between 0.93 and 0.94, suggesting that the sequencing depth was sufficient to obtain the majority of genetic diversity of samples (Table 3). The average number of valid sequences were similar for treatments, being 27,131, 28,812, 26,413 and 30,405, for Control, Urea, Manure and U+M, respectively, with a mean read length of 376 bp. The number of OTUs was not affected by N addition and ranged between 4,072 and 4,295. The indexes of richness and diversity, Chao1, ACE, Simpson and Shannon were also not affected by the treatments.

The addition treatments resulted in an apparent clustering in β-diversity of the soil bacterial community (Fig. 2). Two groups with (Manure and U+M) and without (Control and Urea) manure application occurred along axis PCoA1, with a significant dissimilarity ($P < 0.001$). The PCoA explained 73% of the total variation in the composition of the bacterial community, with PCoA1 and PCoA2 explaining 61% and 12%, respectively. At the bacterial phylum level, the abundance of *Planctomycetes*, *Bacteroidetes* and *Ignavibacteriae* increased with manure additions, whereas that of *Latescibacteria*, *Acidobacteria*, *Armatimonadetes*, *Actinobacteria* and *candidate* division WPS-2 decreased (Table 4). There was no treatment effect on the abundance of *Proteobacteria*. At archaeal phylum level, the abundance of *Thaumarchaeota and Euryarchaeota* decreased with manure application.

Manure and U+M treatments doubled the gene copy number of *AOA* (Fig. 3A). Similarly, manure application also significantly ($P < 0.001$) increased the copy number of the *AOB* gene in manure than non-manure amended treatments. The copy number of *AOA* was generally one order of magnitude greater than that of *AOB*. Further, *AOB* copy

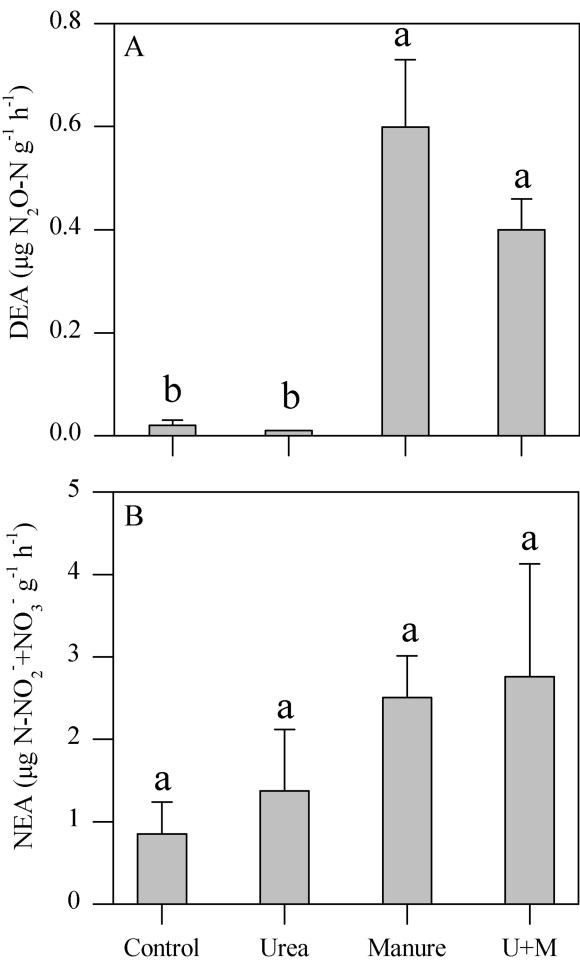

**Figure 1 Denitrifying enzyme activity (DEA, A) and nitrifying enzyme activity (NEA, B) as affected by addition treatments in soil of the drip-irrigated cotton field used in this study.** U+M: 50% urea + 50% manure. Data are means + 1 standard error, $n = 4$. Means followed by the same letter are not significantly different at $P < 0.05$ (Tukey's HSD).

**Table 3 Diversity of the microbial community at a similarity level of 97% for operational taxonomic units (OUT) as affected by addition of treatments in the drip-irrigated cotton field used in this study.**

| Treatment | Coverage index | Reads | OTUs | Shannon | ACE | Chao1 | Simpson |
|---|---|---|---|---|---|---|---|
| Control | 0.93 ± 0.00[a] | 27,131 ± 854[a] | 4,353 ± 100[a] | 7.0 ± 0.12[a] | 7,001 ± 662[a] | 6,354 ± 240[a] | 0.0070 ± 0.0023[a] |
| Urea | 0.94 ± 0.00[a] | 28,812 ± 63[a] | 4,395 ± 70[a] | 6.9 ± 0.07[a] | 7,001 ± 385[a] | 6,429 ± 63[a] | 0.0068 ± 0.0011[a] |
| Manure | 0.94 ± 0.01[a] | 26,413 ± 2,397[a] | 4,072 ± 164[a] | 7.1 ± 0.04[a] | 6,106 ± 147[a] | 5,972 ± 151[a] | 0.0031 ± 0.0002[a] |
| U+M | 0.94 ± 0.00[a] | 30,405 ± 1,280[a] | 4,291 ± 171[a] | 7.1 ± 0.07[a] | 6,413 ± 262[a] | 6,323 ± 278[a] | 0.0031 ± 0.0004[a] |

**Notes:**
Values are means ± 1 standard error, $n = 3$ (For each treatment, soils from three replicated plots were sampled for high-throughput sequencing analysis).
Means within a column followed by the same letter are not significantly different at $P < 0.05$ (Tukey's HSD).

number responded more to treatment additions than that of *AOA*. As a result, manure addition reduced the ratio of *AOA/AOB*, being 88.7, 27.6, 15.8 and 17.0 for Control, Urea, Manure and U+M, respectively.

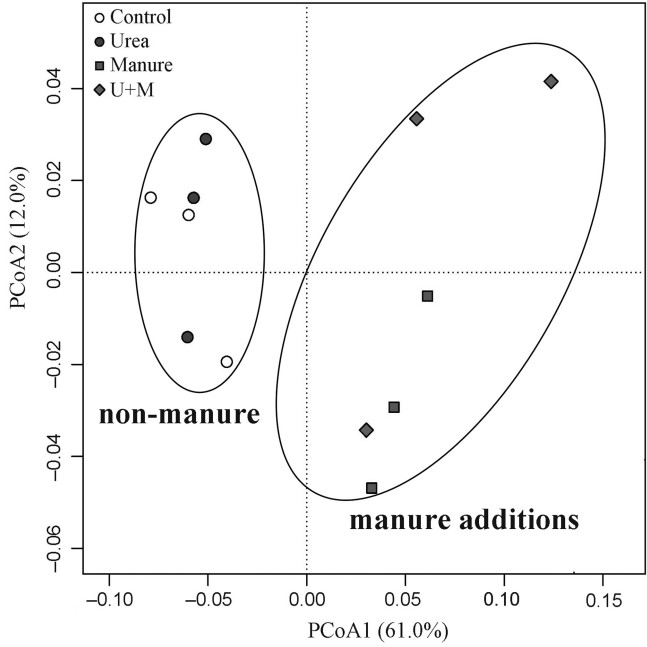

**Figure 2 Principal Coordinates Analysis (PCoA) plot of weighted Unifrac distance matrix showing patterns of β-diversity in microbial communities as affected by the addition treatments to soil of the drip-irrigated cotton field used in this study.** U+M: 50% urea + 50% manure.

Manure addition significantly ($P < 0.001$) increased the copy number of *narG, nirK* or *nosZ* genes, but did not affect that of *nirS* (Figs. 3B–3E). Copy number of *narG* was 27.5–39.0 times greater with manure (U+M and Manure) than non-manure (Control and Urea) addition treatments. Copy number of *nirK* was 3.4–3.7 times greater with manure than non-manure addition treatments. Similarly, the copy number of *nosZ* gene were 9.6-25.2 times greater in manure than non-manure addition treatments.

### Relationships between DEA, NEA, soil properties and microbial abundance

Copy number of nitrifier (*AOB* and *AOA*), nitrate reducer (*narG*), *nirK*, and *nosZ*-type denitrifier genes, but not *nirS*-type denitrifier gene, were positively correlated with $NO_3^-$, DOC, total N, total C and C:N ratio (Table 5). In contrast, soil $NH_4^+$ and pH were not significantly correlated with the copy number of any of the functional genes. There were also significant positive correlations between DEA and abundance of *AOB, AOA, narG, nirK* and *nosZ* genes. NEA, however, was not correlated with the abundance of any functional gene, except for a positive correlation with *nirK*.

The bacterial β-diversity was highly associated with changes in soil environmental variables (Fig. 4). The first and second axes explained the variance in bacterial community structure to 38.6% and 13.9%, respectively. The bacterial community of manure addition treatments (Manure and U+M) was mainly associated with soil concentrations of $NO_3^-$, DOC, total C, total N and C:N ratio. In contrast, the bacterial community of treatments without manure addition (Control and Urea) was mainly associated with soil pH.

**Table 4 Relative abundances (%) of selected bacterial and archaeal taxa at the phyla level as affected by addition of treatments to soil in this study.**

| | Bacteria | | | | | | | | | Archaea | |
| --- | --- | --- | --- | --- | --- | --- | --- | --- | --- | --- | --- |
| | Proteobacteria | Planctomycetes | Acidobacteria | Actinobacteria | Bacteroidetes | Armatimonadetes | Ignavibacteriae | Candidate division WPS-2 | Latescibacteria | Thaumarchaeota | Euryarchaeota |
| Control | 32.5[a] | 8.9[b] | 11.5[a] | 11.6[a] | 5.6[b] | 0.62[a] | 0.13[b] | 0.07[a] | 0.12[a] | 0.58[a] | 0.77[a] |
| Urea | 32.2[a] | 8.3[b] | 10.7[a] | 12.1[a] | 6.2[b] | 0.67[a] | 0.14[b] | 0.04[a] | 0.08[ab] | 0.50[ab] | 0.87[a] |
| Manure | 34.6[a] | 11.1[a] | 8.8[b] | 9.0[b] | 11.4[a] | 0.37[b] | 0.25[a] | 0.01[b] | 0.06[b] | 0.24[bc] | 0.24[b] |
| U+M | 35.1[a] | 10.0[a] | 7.5[b] | 9.4[b] | 8.3[ab] | 0.27[b] | 0.20[ab] | 0.01[b] | 0.04[b] | 0.30[b] | 0.24[b] |

**Note:**
Means followed by the same letter are not significantly different at $P < 0.05$ (Tukey's HSD).

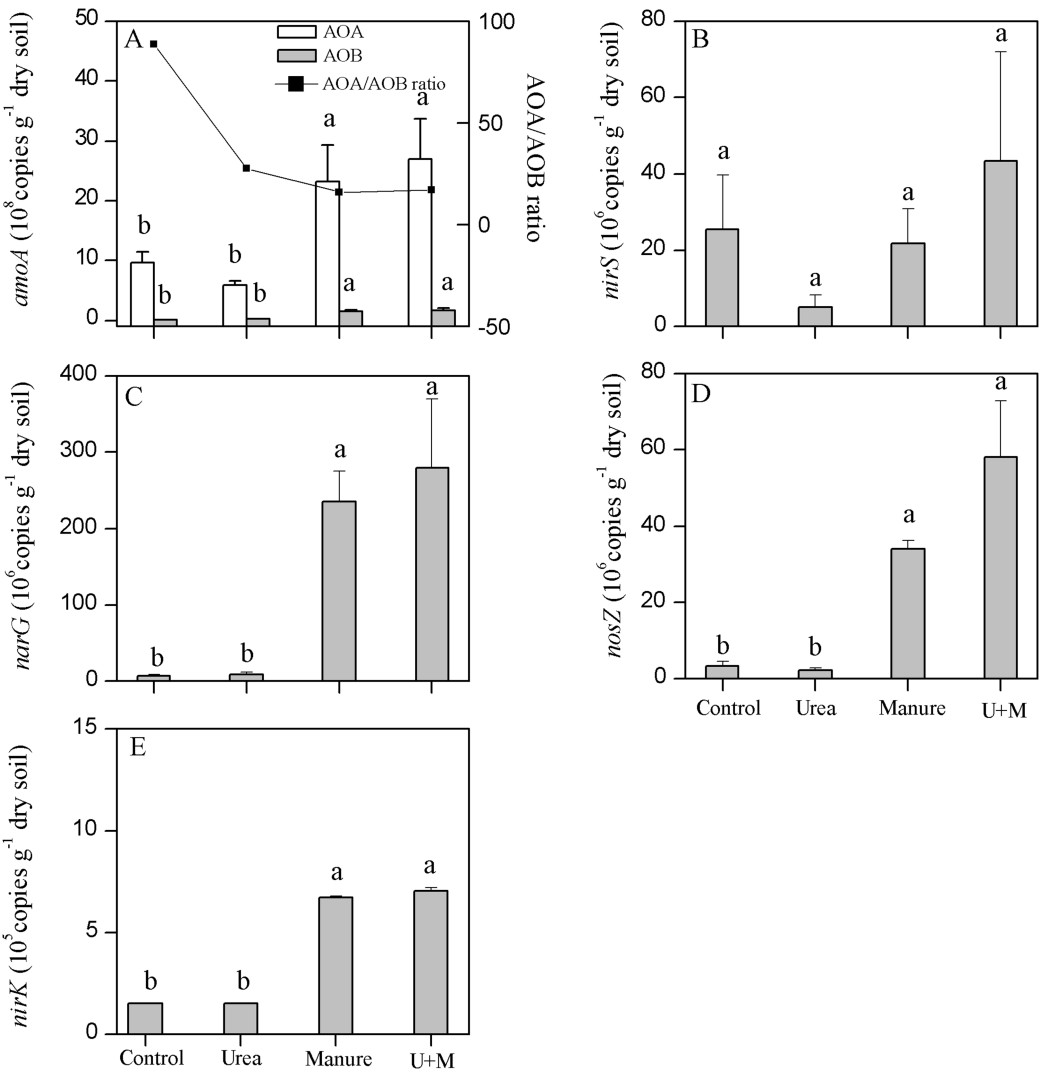

**Figure 3 Copy numbers of archaeal (*AOA*) and bacterial (*AOB*) *amoA* (A), *nirS* (B), *narG* (C), *nosZ* (D) and *nirK* (E) genes in soil as affected by the addition of treatments to plots in the drip-irrigated cotton field used in the study.** Data are means + 1 standard error, *n* = 4. Means followed by the same letter are not significantly different at *P* < 0.05 (Tukey's HSD).

## DISCUSSION

Manure application exerted a significant effect on microbial abundance and beta diversity and greatly increased the DEA compared with conventional urea. We further linked the increase of DEA by manure application with changes in denitrifier abundance. The increased denitrification activity with manure application was in accordance with the increasing abundance of nitrate reducer (*narG*), and *nirK*- or *nosZ*-type denitrifiers. It should be noted, however, soil samplings were conducted for only one time over the growing season for determination of soil microbial activities in the current study, which hindered the investigations of temporal changes in soil microbes and could also cause uncertainties in correlating with $N_2O$ emissions. Still, sampling was done in a

**Table 5 Pearson correlation coefficients (r) between copy number of N$_2$O-related functional genes and soil characteristics across all treatments and replicates, n = 16.**

|  | AOA | AOB | narG | nosZ | nirS | nirK |
|---|---|---|---|---|---|---|
| NO$_3^-$ (mg kg$^{-1}$) | 0.65** | 0.61* | 0.81*** | 0.65** | 0.24 | 0.77*** |
| NH$_4^+$ (mg kg$^{-1}$) | −0.29 | 0.15 | −0.58 | −0.14 | −0.24 | 0.01 |
| DOC (mg g$^{-1}$) | 0.70*** | 0.73*** | 0.86*** | 0.76*** | 0.47 | 0.73*** |
| pH | −0.22 | −0.30 | −0.40 | −0.25 | −0.28 | −0.33 |
| TN (g kg$^{-1}$) | 0.63* | 0.71** | 0.66** | 0.58** | 0.21 | 0.63** |
| TC (g kg$^{-1}$) | 0.85*** | 0.82*** | 0.82*** | 0.72** | 0.31 | 0.76*** |
| C/N | 0.72** | 0.61* | 0.67** | 0.58* | 0.29 | 0.62** |
| DEA (ug N$_2$O-N g$^{-1}$ h$^{-1}$) | 0.57* | 0.85*** | 0.84*** | 0.70** | 0.31 | 0.76*** |
| NEA (ug N-NO$_2^-$+NO$_3^-$ g$^{-1}$ h$^{-1}$) | 0.19 | 0.26 | 0.18 | 0.06 | −0.22 | 0.64** |

**Note:**
\* \*\* \*\*\* Indicate significance at $P < 0.05$, $< 0.01$ and $< 0.001$, respectively.

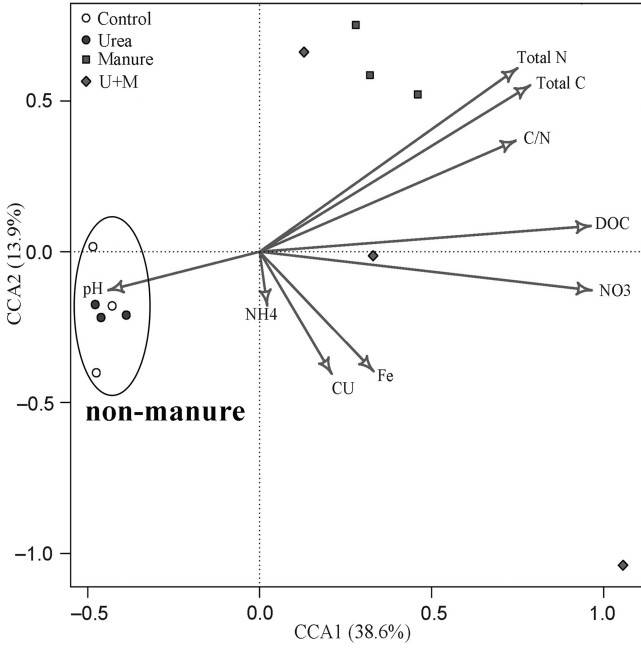

**Figure 4 Canonical correspondence analysis (CCA) bi-plot of soil properties in relation to microbial OTUs as affected by the addition of treatments to soil of the drip-irrigated cotton field used in this study.**

representative field for the local cotton production where we compared farmer's management practices of applying manure relative to inorganic fertilizers. The sampling depth (0–20 cm) for microbial analysis was also in accordance with previous findings that soil N$_2$O emissions following N addition were mostly attributed to the topsoils (*Wagner-Riddle et al., 2008*; *Kuang et al., 2019*).

## Impact of N addition strategy on denitrification and nitrification

The increased activity of soil denitrifying enzymes with manure in the current study is inconsistent with our findings at the same field where we reported more N$_2$O emissions

from manure compared with urea application under drip irrigation conditions (*Kuang et al., 2018*). The increased $NO_3^-$ and carbon supply with manure application could likely have provided the primary substrate for denitrification and increased the $N_2O/(N_2O + N_2)$ ratio (*Francis et al., 2013*). *Chantigny et al. (2010)* also suggested that manure can elevate soil respiration and deplete $O_2$ concentration to create temporary anaerobic conditions, thereby further increasing the proportion of $N_2O$ production through denitrification. These studies highlight the importance of N addition source on soil N transformation processes and suggest that manure induced $N_2O$ emissions are likely attributed to denitrification.

In contrast to DEA, NEA was not affected by manure application in the current study. Similarly, *Shen et al. (2008)* also reported that organic manure did not affect potential nitrification rates of an alkaline sandy loam soil in northern China. Several studies suggested that soil pH is the dominant factor for nitrification as it determines the availability of $NH_4^+$, which is the primary substrate for ammonia oxidation, the initial and rate-limiting step of nitrification (*Fan et al., 2011*; *Nicol et al., 2008*). In our study, both pH and the availability of $NH_4^+$ were not affected by N addition strategy, confirming the insensitivity of NEA to N sources.

In contrast with manure, urea did not significantly affect DEA and NEA compared to Control. Our results agree with those of *Yin et al. (2015)*, who reported that manure but not inorganic fertilizer increased denitrification potential. In contrast, application of inorganic N fertilizers increased the activity of nitrification (*Fang et al., 2018*; *Shi et al., 2016*) and potential denitrification (*Duan et al., 2017*; *Wang et al., 2018*). The absence of the inorganic fertilizer effect in the current study was associated with the minor to no effect by urea application on soil properties such as pH, DOC and inorganic N ($NO_3^-$ and $NH_4^+$) compared with Control. It is likely the buildup of C and N substrates by urea application were not sufficient enough to affect the activities of functional genes.

## Impact of N addition strategy on the abundance of functional genes and bacterial community structure

In the current study, the positive relationships of the abundances of *narG*, *nirK* and *nosZ* with DEA and further with soil DOC, total C and total N suggest that manure significantly increased gene abundance by providing C and N substrate. This result is in line with the previous findings that the denitrifiers abundance could be used as a predictor of DEA (*Morales, Cosart & Holben, 2010*). Our findings also agree with previous studies which reported that organic manure increased the abundance of *nosZ*-type denitrifier compared to inorganic fertilizers (*Hallin et al., 2009*; *Tao et al., 2018*). Also being consistent with previous studies (*Zhou et al., 2011*), the abundance of *nirK* but not *nirS* was increased by manure application in this study, suggesting that *nirK* was more susceptive to fertilizer regimes than *nirS*-type denitrifier. *Hallin et al. (2009)* also reported that denitrification rates were not correlated with the abundance of *nirS* genes in soils treated with different fertilizer regimes for 50 years. A possible reason for the lack of correlation could be that the denitrifier harboring the *nirS* gene might play a minor functional role for DEA (*Attard et al., 2011*). In the current study, the nitrate reducer (*narG*), and *nirK*- and

*nosZ*-type denitrifiers, which encodes the main catalytic enzymes responsible for nitrate reduction, nitrite reduction, and $N_2O$ reduction respectively, were more sensitive to manure application. The increase of the denitrifiers abundance with manure application thus increased the pool of denitrifying enzymes. Even though the limited soil sampling for microbial analysis hindered the possibility of directly linking results from the current study to the in-situ measurements of $N_2O$ flux, the positive relationship between DEA and the abundance of denitrifiers suggest the manure-induced $N_2O$ emissions in *Kuang et al. (2018)* was more likely determined by denitrification.

It is interesting to note that manure application increased *AOA* and *AOB* whereas did not affect NEA in this study, suggesting the abundance of ammonia-oxidizers are not necessarily associated with nitrification potential. *Nicol et al. (2008)* reported that the activity of ammonia-oxidizers was more associated with the relationships among transcription, translation, and enzyme function rather than the abundance of functional genes. It is also likely that the complicated subsequent hierarchical regulation of enzyme expression resulted in an uncouple effect between NEA and *amoA* gene abundance (*Röling, 2007*). Consistent with previous studies (*Fan et al., 2011*; *Tian et al., 2014*), N addition reduced the *AOA/AOB* ratio in this study, suggesting that *AOA* and *AOB* may occupy different soil niches due to the differences in physiological and metabolic pathways. Previously, *AOA* prefers low $NH_3$ substrate conditions for growth, whereas *AOB* prefers higher $NH_3$ levels (*Di et al., 2010*), thus potentially resulting in a lower *AOA/AOB* ratio following N addition.

Similar to previous studies (*Ji et al., 2018*; *Kumar et al., 2018*; *Wang et al., 2019*), manure application significantly changed β-diversity of soil bacterial community in the current study. The PCoA analysis revealed a dominant contribution of PCoA1 (61%) to total variation and a clear separation of manure vs. non-manure groups along the axis PCoA1. This suggests that the addition of manure was a key factor determining the variation in the bacterial community among treatment. Clearly, the increased N and C substrates with manure application have increased the growth of some specific microbial groups and suppress others and thus changed the composition of the soil microbial community. The absence of urea effect on the β-diversity of the bacterial community was attributed to the low organic matter content (6.6 g kg$^{-1}$), suggesting that the substrate deficiency of C limited microbial activities under the conditions in this study. In this study, manure or urea applications did not influence α-diversity of soil bacterial community, likely due to an absence effect on soil pH. *Fierer & Jackson (2006)* reported that soil pH is the primary driver determining the α-diversity and richness of the soil bacterial community.

In the current study, the changes of soil bacterial community structure in response to manure application were attributed to the increasing relative abundance of *Planctomycetes*, *Bacteroidetes*, *lgnavibacteriae* and decreasing abundance of *Actinobacteria*, *Acidobacteria*, *Latescibacteria*, *Armatimonadetes* and *candidate* division WPS-2. These results highlight the change of eutrophic and oligotrophic bacteria. For example, *Fierer, Bradford & Jackson (2007)* found that *Bacteroidetes* were typically copiotrophic bacteria and could thrive in soil with high available organic carbon. *Planctomycetes* are involved in the turnover of soil organic carbon, and nutrient availability, and the reproduction of

this microbial group may increase intensively in response to the application of manure (*Lupatini et al., 2016*). The phyla which were negatively influenced by manure application were considered as slow-growing oligotrophs accustomed to nutrient-limited environments. For example, several studies had shown that *Acidobacteria* strains grew slowly with their growth being limited to substrate additions (*Goldfarb et al., 2011*).

## CONCLUSIONS

Manure application significantly increased the abundances of nitrate reducer (*narG*), and *nirK-* and *nosZ*-type denitrifier gene abundances and DEA. Additionally, soil DOC, total C and total N contents increased with the abundances of *narG*, *nirK and nosZ* genes, suggesting manure addition to soil stimulated production of these functional genes by providing energy and N substrate. In contrast, urea application did not affect the abundances of nitrifier and denitrifier functional genes. High throughput sequencing clearly showed that 2 years of manure application significantly altered the bacterial community composition of soil. Consequently, our study demonstrated a strong link between abundances of nitrate reducer (*narG*), *nirK-* and *nosZ*-type denitrifier gene copies and enhanced DEA with manure application under drip-irrigation of cotton. Together the results indicate denitrification was likely the key process leading to a manure-induced increase in $N_2O$ emissions.

### Funding

This study was funded by National Natural Science Foundation of China (No. 31570002, 31870499) and China scholarship Council (NO. 201804910570) and the Poverty Alleviation Program of Chinese Academy of Sciences (KFJ-FP-201903). The funders had no role in study design, data collection and analysis, decision to publish, or preparation of the manuscript.

### Grant Disclosures

The following grant information was disclosed by the authors:
National Natural Science Foundation of China: 31570002, 31870499.
China scholarship Council: 201804910570.
Poverty Alleviation Program of Chinese Academy of Sciences (KFJ-FP-201903).

### Competing Interests

The authors declare that they have no competing interests.

### Author Contributions

- Mingyuan Yin conceived and designed the experiments, performed the experiments, analyzed the data, prepared figures and/or tables, authored or reviewed drafts of the paper, approved the final draft.
- Xiaopeng Gao conceived and designed the experiments, analyzed the data, authored or reviewed drafts of the paper, approved the final draft.
- Mario Tenuta conceived and designed the experiments, authored or reviewed drafts of the paper, approved the final draft.
- Wennong Kuang performed the experiments, prepared figures and/or tables, approved the final draft.
- Dongwei Gui performed the experiments, contributed reagents/materials/analysis tools, authored or reviewed drafts of the paper, approved the final draft.
- Fanjiang Zeng conceived and designed the experiments, contributed reagents/materials/analysis tools, authored or reviewed drafts of the paper, approved the final draft.

## Data Availability

The raw measurements are available as a Supplemental File.

## Supplemental Information

Supplemental information for this article can be found online at http://dx.doi.org/10.7717/peerj.7894#supplemental-information.

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
