# Peer review of "Manure application increased denitrifying gene abundance in a drip-irrigated cotton field"

_PeerJ, doi:10.7717/peerj.7894_

## Round 0.1 · original submission · Major Revisions

The manuscript – Manure application increased denitrifing gene abundance… - received missed reviewers. Two reviewers had serious concerns. One reviewer rejected the manuscript because of soils were only sampled once and the number of replicates was low. This approach could mix the peak fluxes responsible for the bulk of the N2O release. I share these concerns but, as I understand it, this paper is a companion paper to an earlier work (Kuang et al. 2018). I presume it presents a more comprehensive sampling regime that captures the peak fluxes. I suggest you discuss how the samples analyzed in the PeerJ submission are representative and sample the community responsible for the bulk of N2O release.

The abstract does not state a hypothesis. An earlier paper from your group states that manure application increased N2O release by influencing both nitrification and denitrification (Kuang 2018). The title of this manuscript suggests the hypothesis that it is denitrifiers. Provide background in Introduction and then state a hypothesis in Abstract and Introduction.

Given the extent of revisions required and that we received one recommendation to reject, I will need to find a new reviewer to evaluate a resubmission. If you choose to resubmit, address all three of the reviewers concerns and my specific comments.

Specific

Line 24. Delete empty phrases like “Results showed that..”, “It was found (line 314)”, and “has been shown to (line 378).
Line 43. Replace “cause” with “contributes to,” as CFCs and other pollutants contribute to ozone depletion.
Line 46-48. It is not clear that manure application contributes significantly to the global production of N2O. Revise to state that “Agricultural fields fertilized with manure produce more N2O than field fertilized with synthetic fertilizes…”
Line 56. Here and throughout, avoid the passive voice. Revise to “NH2OH limits nitrification”, “..pH, controlled nitrification…(line 65),” “Specific reductases..regulate each step…(line 72)”, “…reported that application of synthetic fertilizer reduced bacterial diversity…(line 95)”
Line 67. N-cyclying prokaryotes, not functional gene groups, control activity.
Line 79. Replace “old” with “long”
Line 91. Delete “with manure”
Line 94-96. Revise to “…reduced bacterial diversity”
Line 177. Revise to “defrosted”
Line 261. Delete “Richness..Table 3.”
Line 279. Delete “Real-time..genes.”
Line 310. Delete “assessed…under drip irrigation …and”
Line 403. Delete “Further studies..processes”
Line 407. Delete “This study…We found that..”

[]

Reviewer 1 ·

Basic reporting

Overall, the article is well written and clear. However, there is a little spelling and format mistakes.

L30, “(r=0.70-084, P<0.01)” should have space between letter, character, and value, “(r = 0.70 - 084, P < 0.01)”, please check the whole article.
L40, “Denitrifying enzyme activitie” should be “Denitrifying enzyme activities”.
L47, “inorganic” better to be “inorganic sources”.
L99, “NW China” better to be “northwestern China”.
L134, “The used manure” should be “The manure used for this study”.
L175, “Santa Clara, CA” should be “Santa Clara, CA, USA”, please consistently list the machine name, company name, company location (city, state, country), or part of these information, but should be consistent in whole article.
L247, “Manure application increased soil total N content” better to be “Manure application (both Manure and U+M treatments) increased soil total N content”.
L262, L287, please consistently use comma before “and”, for example, “being 88.7, 27.6, 15.8, and 17.0 for Control, Urea, Manure and U+M, respectively.” should be “being 88.7, 27.6, 15.8, and 17.0 for Control, Urea, Manure, and U+M, respectively.” Please check the whole article.
L365, “the nitrate reducer (narG), and nirK and nosZ-type denitrifiers” should be “the nitrate reducer (narG), and nirK- and nosZ-type denitrifiers”.
L387-388, “The absence of urea treatment on microbial community was not unexpected as the soil was low in organic matter of only 6.6 g kg-1”, not clear, please reorganize this sentence.
L417, “nirK- and -nosZ type denitrifiers” should be “nirK- and nosZ-type denitrifiers”
L428 & L432, consistent in format, “Lewis Publishers, Boca Raton, FL” or “Lewis Publ. CRC Press, Boca Raton, Florida”. Please check all references, make sure in uniform format.
Table 3, “Chao” should be “Chao1”
Table 5, “*, **, *** indicate significance at P < 0.05%, < 0.01% and < 0.001%, respectively.” should be “*, **, *** indicate significance at P < 0.05, < 0.01 and < 0.001, respectively.”
Fig. 3, better to increase the spacing between columns

The author provided sufficient information and all the information the author provided are clear and sticks to the main idea. For example, the authors provided sufficient introduction and background, in order to state the research question and objective. However, it is better to provide bacteria or fungi species when cited prior studies, which can provide some information about the richness and diversity of soil microbial community so that match the findings (Fig. 2 and Table 3&4). In addition, all prior literature were appropriately referenced.

L238, “(Oksanen et al., 2011)”, I didn’t find it in reference section, the one I found is “Oksanen, 2011”, it should be “(Oksanen, 2011)” or add the correct one.

The article is well organized, especially the introduction and experimental design sections, which is eligible and well organized. And the hypothesis and objectives are well matched to each other, which is clear for us to read. All the tables and figures are relevant the content of the article, and appropriately described and labeled. And all raw data are appropriate and available in accordance.

Experimental design

The subject that is appropriate for publication in PeerJ. The author clearly state the research question and list the objective, which is meaningful. There is an ever increasing body of knowledge published in scientific literature, which showed application of organic manure increased N2O emission under greenhouse and filed condition. However, litter information about how application of organic manure and inorganic fertilizer effect on the gene abundances and activity of nitrifier and denitrifier communities under drip irrigated conditions. This article indicated that manure application increased the abundances of denitrifing functional genes and denitrification potential, which could contribute to the increased N2O emissions.

The materials and methods used are appropriate and no repeat information. While, some detail information were missing.

L119, the author gave the mean annual air temperature, it’s better to provide the air temperature during the growing seasons.
L129, should provide more plot information, buffer separating plots within block and buffer separating the blocks.
L130, please provide the available N content for urea and manure. For example, urea (46% N).
L136, the N, P, K content of manure based on dry basis or wet basis, what is the moisture of the manure, and what is the pH of the manure?
L140, cotton seed from what company, and what is the planted density of cotton seeds?
L235 & L237, the Vegan package for PCoA and CCA, are they in the same version? If it is, please move the Version and reference from L237 to L235.

Validity of the findings

Overall, the findings are valid and meaningful. All results and conclusion are well stated and based on the data. All underlying data are provided and analyzed by appropriate statistical methods. The conclusion are matched the research question (hypothesis and objectives) and indicated the limitation of this study and future questions needed to investigate.

L251-252, “soil C:N ratio were 37-100% greater in Manure and U+M than in Urea and Control treatments”, while no significant difference between Manure and Control treatments, better to change this sentence, separate Manure and U+M into two sentences, or just keep U+M, for example “soil C:N ratio were 48-100% greater in Manure than in Urea and Control treatments”.
L270-274. PCoA1 explained 61% of the total variation and the microbial communities under different fertilizer treatments were separated into two groups along axis PCoA1, so it is better to provide what features contribute to PCoA1.
L300-301, “NEA, however, was not correlated with the abundance of any functional gene” is not consistent to Table 5, which showed that NEA was positively correlated to nirK.

Reviewer 2 ·

Basic reporting

No comment

Experimental design

The objective stated in the abstract should be improved. As it stand it is not clearly indicated that this study also evaluated the diversity of bacterial and archaeal communities in this system.

There was a single date in one year used to assess the effect of fertiliser sources on the abundance of nitrifier and denitrifier and bacterial-archaeal communities. Given that nitrification and denitrification are episodic and that environmental conditions affect both abundance, microbial diversity and denitrification and nitrification activity, the use of a single date to understand this system is insufficient.

The relationships between the abundance of nitrifier and denitrifier and bacterial-archaeal communities and DEA and NEA were explored but not with actual N2O emissions from this system. DEA and NEA are measurements of the potential denitrification and nitrification but it is not the measurements of the actual N2O emissions of this system. N2O emissions were measured using static chambers in Kuang et al 2018. It is unfortunate that the authors did not try to use this data and explore correlation with the abundance of denitrifier and nitrifiers with actual.

There is insufficient details and rationale on the crop production system including if this is short term or long term effect of fertilizer application that is under study, the crop production system and practice (planting date).

Validity of the findings

The discussion is too speculative with inferring that the results from this study explains in situ N2O emissions previously published under Kuang et al. 2018. DEA and NEA are measurements of potential denitrification and nitrification so it should not be interpreted as the actual measurement of these processes.

Additional comments

Specific comments
Line 25: Microbial is vague. Is this bacterial or fungal communities?
Line 29-34: The results in the abstract need to be presented more systematically. The changes in the abundance of denitrifiers, nitrifiers and nitrate reducers in this system should be first described. The possible relationships between the abundance of denitrifiers, nitrifiers and nitrate reducers with DEA, NEA and soil properties should then be described.
Line 29-31: This sentence does not make sense: Real-time quantitative PCR’ does not reveal an increased in DEA (please remove ‘Real-time quantitative PCR’), nor is the correlation with increased abundance (remove ‘increased’).
Line 31: denitrifiers
Line 47: inorganic fertilizer. Furthermore, please be consistent in the term used in the study, either used inorganic, mineral or synthetic fertiliser. I suggest to select one and change throughout the manuscript.
Line 71: It is correct that both denitrifiers and nitrate reducers can reduce nitrate to nitrite but here you are specifically stating the steps for denitrification so I suggest removing the nitrate reducers from this sentence.
Line 74-77: this information is correct but irrelevant to the study since fungal denitrifiers are not studied so I suggest removing.
Line 101: water and nutrient use efficiency by the crop
Line 126: when did manure application starts in this system? Basically, are we seeing the short-term or long-term effect of application of manure and inorganic fertilizer?
Line 134: remove ‘used’
Line 147: Why was a single date sampled? What is the rationale for choosing this particular date? It would have been logical to use a date at which N2O emissions were measured in the Kuang et al 2018 paper.
Line 152: what was the rational to freeze the soils for DEA and NEA measurements? These measurements are measuring enzyme activity and it is less than ideal to freeze the samples since it would affect their activity.
Line 168: change defrost for thawed. Explain how the samples were thawed i.e. what temperature and for how long.
Line 190: use quantitative PCR not real-time PCR
Line 195-196: There is not enough information as to what genes were cloned as standard. Is this from specific denitrifier and nitrifier species or an environmental sequence? Difference in the performance of the standards can be observed, given that it is important to be specific as to what was used.
Line 199-201: The range of PCR efficiency, the slope and the intercept should be stated for each PCR reactions.
Line 203: Was the number of sequences normalized across all samples? If not, there would be no way to know if differences among treatments are due to the treatments or the unequal amount of sequences across the samples.
Furthermore, the singletons and low abundance OTUs should be removed to avoid spurious OTUs as suggested by Bokulich et al. 2013.
Line 240-242: This phrase makes no sense (abundance of categories between treatment?), please rephrase.
Line 242-243: corrected p-values of what? Welch’s t-test?
Please use phyla for plural or phylum for singular.
Table 4: The Proteobacteria is among the three most abundant bacterial phyla thus it should be presented.
Line 313: change ‘communication’ to ‘abundance and diversity’

Line 316-318: That DEA is correlated with the abundance of denitrifiers on a single date in this study is not evidence that denitrification was the major process involved at the field level in Kuang et al. 2018. The statement should be removed.
Line 332-334: this statement is irrelevant: this is about findings reported under Kuang et al. 2018 and it should be removed.
Line 342: compared to what? The control?
Line 344: what does denitrification means here, N2O emissions or denitrification rate or DEA? Please make sure that specify this and that you are not comparing with different measurement. DEA is a potential, not the actual measurement.
Line 383: why would they nirS play a more minor role for DEA compared to other denitrifier groups?
Line 367-368: This is incorrect: DEA is not more associated with nitrate, nitrite and N2O. What was measured was the abundance of denitrifiers in soils, while the DEA measured the enzymatic activity under optimal conditions i.e. with carbon and nitrate addition. The conclusion here is that the abundance of denitrifiers has increased with manure application to soils and that also increased the pool of denitrification enzymes in soils so DEA increased too.
Line 381: change Agreement with to similarly to
Line 382: please be specific, it is the bacterial and archaeal community that has been studied. Also what are you referring to alpha or beta-diversity, or relative abundance of phyla?
Line 384: remove reproduction
Line 388: ‘suggesting that C other than N’ – what does this means?
Line 390: this statement contradict the statement made a the start of this paragraph.
Line 393: this is the relative abundance, it is important to mention this.
Line 408: potential NEA and DEA

Reviewer 3 ·

Basic reporting

The article has adequate literature references and the authors clearly state the research problem and objective in the introduction. The results are well presented using appropriate figures and tables for the types of data.

The manuscript should undergo editing for proper English.

The versions of bioinformatics software, including PRINSEQ, UCHIME, and mother are not specified. The software also needs to be cited.

The conclusion states that pyrosequencing was the sequencing technology used, however, the materials and methods clearly state MiSeq sequencing.

Experimental design

Three replicates for each treatment is the absolute minimum, this was perhaps done due to the costs of sequencing. Fortunately they were composites of 4 soil samples but in the future consider including more replicates.

More details regarding the experimental methodology would be beneficial. More specifically:
How was the auger cleaned between samples? How much soil was used for DNA extraction?

Details on the analysis of sequencing data is also lacking. Please elaborate on the parameters used for sequence quality assessment control with PRINSEQ. If defaults were used, specify.

What is the rationale for performing 5 PCR cycles with the 45 ° C annealing temperature prior to 20 cycles with the typical 55 °C temp.?

Line 188: In the future, consider storing DNA TE (or Tris with low EDTA for sequencing applications) as it is a buffered solution and EDTA protects against nucleases.

narG and nirK have very low qPCR efficiencies; is this the case for other studies also? This should probably be addressed.

Validity of the findings

The qPCR results are concerning as no internal amplification control was used to detect PCR inhibition. As such, it is unknown if the difference between organic and inorganic N treatments is due to greater inhibitors in the organic sample or due to an actual difference in gene abundance. Inhibition should be addressed.

In line 385-386, the authors state “It is also likely that manure application had caused high loads of exogenous microbes and thus resulted in the community disturbance”. There is no question of this; manure is loaded with microbial activity. This statement should be clarified to reflect this.

Richness and diversity were not impacted by any of the treatments. Given what we know about soil microbial community structure and organic versus inorganic N addition, this is surprising and contrary to previous studies. This should be explored in the discussion as it is presently not mentioned.

As there was no detected difference between the number of OTUs across any treatment, is it possible the 97% species identity cutoff was too low for the V3-V4 region of the 16S rRNA gene? Please justify the cutoff and consider rerunning with a greater cutoff. Conversely, it is possible that inadequate parameters were used in PRINSEQ resulting in error prone reads being included in the analysis and obscuring differences between treatments. Please include the parameters used for PRINSEQ.

---

## Round 0.2 · Minor Revisions

We received one review of accept for the resubmitted manuscript – Manure application increased denitrifing gene abundance…; however, I believe the manuscript still needs revision. The comments I provided to the first revision were not addressed adequately. Importantly, abstract still does not state a hypothesis and there are a number of edits for style (see below) that must be addressed.
Specific
Line 17. Between “…manure can increase N2O emission” and “We tested…” state the hypothesis, which I think is that there is a link between N2O flux and the bacterial community structure.
Line 18. You also measured NEA and DEA. You need to state that enzymatic activity was measured too.
Line 19-22. Revise to “… in an arid…” and delete “to investigate..diversity of the bacterial community.”
Line 24. Revise to “Manure was broadcast..”
Line 26. Revise to “…did not, as assessed by nextgen sequencing of PCR-amplicons generated from rRNA genes, effect alpha diversity of bacterial communities but changed beta diversity..”
Line 38. Something is missing at the end of Abstract. I suggest revising to “… and denitrification potential. This suggests that manure application increases N2O emission by increasing denitrification and the population of bacteria that mediate that process.
Line 44. Provide reference for the contribution of N2O to global warming and ozone depletion.
Line 48. Avoid phrases like “many studies have reported that…” and “Previous studies suggested that (line 75).” Revise to “Manure application results in more..”
Line 59. Delete “through their effects on soil properties”
Line 66. Revise to “…study of fertilized forest soils in China found that..”
Line 75. Here and throughout, avoid the passive voice. Revise to “Changes in the abundance and community of denitrifiers can largely explain the increase in denitrification associated with fertilizer application…”
Line 81-84. Delete “The inconsistent..denitrification activity.”
Line 85. Delete “including inorganic fertilizer and animal manure”
Line 87. Delete “is a primary..and its”
Line 96. Revise to “..production receives intensive”
Line 97-98. Delete “due to ..by the crop.”
Line 270-272. Delete “DEA values..Control Treatments.” Revise to “Manure and U+M treatment significantly increased DEA levels… In contrast, NEA levels did not respond significantly to fertilizer treatments.
Line 294. As above, start each paragraph in Results, with a statement of the key result. In this case. “Manure and U+M treatments doubled the copy number of AOA..”
Line 325. I thought manure application did not effect alpha diversity.
Line 402. Revise to “PCoA1. This suggests that addition of manure…”

Reviewer 1 ·

Basic reporting

no comment

Experimental design

no comment

Validity of the findings

no comment

Additional comments

Thank you for responding to my comments. I am recommending acceptance of the manuscript to the technical editor.

---

## Round 0.3 · Minor Revisions

The manuscript is acceptable in terms of content; however, it needs close review by all authors for format and style. I suggest you ask a colleague to proofread it. I provide some points below from the first 100 lines but this should not be considered a comprehensive list.

Line 17. The phrase “with manure” doesn’t make sense and it is self-evident that enzyme activity is linked to flux. I suggest revising to “We tested the hypothesis that increase in N2O flux from soils fertilized with manure reflects a change in bacterial community structure and, specifically, an increase in the number of denitrifiers. To test this hypothesis, a field experiment…”
Line 23. Replace “addition” with “fertilizer”
Line 26. Revise to “…increased the abundance of genes associated with nitrate reduction (narG) and denitrification (nirK and nosZ).”
Line 29. Delete “compared to..treatments.”
Line 30. Pluralize to “genes”
Line 31. Delete “with manure application”
Line 32. Revise to “…with the abundance of the genes narG, nirK and nosZ.
Line 42. For use “et al.” for more than two authors.
Line 46. Present published work in present tense. Revise to “Manure application increases N2O emissions more than..”
Line 47-48. The previously published work (2014 and 2017) doesn’t confirm a 2018 paper. Revise to “…Zhou et al. 2017). We recently observed that manure application by drip irrigation results in particularly high fluxes (Kuang et al. 2018); however, it..”
Line 56-57. Revise to “N drives the abundance…”
Line 65-67. Delete “Overall..drip irrigation.”
Line 77-79. Revise to “…total N, determine the rate of denitrification.”
Line 88. Delete “a” in “a decreasing”
Line 476 and 482. Proofread references. Replace “Isme” with “ISME”
Line 502. Be consistent in Journal titles, either abbreviate all or not. “Frontiers in Microbiology” is not abbreviated on the same page (Lines 465 and 474).
Line 543. Italicize journal tittles

---

## Round 0.4 · accepted · Accept

Xiaopeng,

The manuscript is much improved and acceptable for publication. I think this work will further our understanding of the link between the structure of denitrifier communities and the fluxes of the processes they mediate, which interests me personally and should interest the community.

I have some minor comments to consider in production, which I will share with the editor.

Line 17. Revise to “increased”
Line 47. Delete “and”
Line 51. Delete “that lead to N2O production”

Regards,

Michael